# IS A SMALL MATRIX EIGENDECOMPOSITION SUFFICIENT FOR SPECTRAL CLUSTERING?

## ABSTRACT

Spectral clustering has been widely used in clustering tasks due to its effectiveness. However, its key step, eigendecomposition of an $n \times n$ matrix, is computationally expensive for large-scale datasets. Recent works have proposed methods to reduce this complexity, such as Nyström method approximation and landmark-based approaches. These methods aim to maintain good clustering quality while performing eigendecomposition on a smaller matrix. The current minimum matrix size for spectral decomposition in spectral clustering is $k \times k$ (where $k$ is the number of clusters). However, no existing algorithms can achieve good clustering performance with only a $k \times k$ matrix eigendecomposition. In this paper, we propose a novel distribution-based spectral clustering. Our method constructs an $n \times k$ bipartite graph between $n$ data points and $k$ distributions, enabling the eigendecomposition of only a $k \times k$ matrix and preserving clustering quality at the same time. Extensive experiments performed on synthetic and real-world datasets demonstrate the superiority and effectiveness of the proposed method compared to the state-of-the-art algorithms.

## 1 INTRODUCTION

Clustering is an unsupervised learning method that does not require labeled data, so it plays an important role in many fields where labeling is difficult. Spectral clustering (Von Luxburg, 2007; Shi and Malik, 2000), as one of the most widely used clustering algorithms, has solid theory and good clustering performance, and has been applied to many fields, such as image segmentation (Li et al., 2012), cell analysis (Zu et al., 2023), and multi-view clustering (Lu et al., 2022; Tang et al., 2022).

Spectral clustering has two key steps: constructing a similarity matrix and eigendecomposition, which have time complexities of $\mathcal{O}(n^2)$ and $\mathcal{O}(n^3)$ ($n$ is the number of points) respectively (Von Luxburg, 2007; Li et al., 2022). The expensive time complexity limits the application of spectral clustering in processing large-scale data (Li et al., 2022; Huang et al., 2019; Xie et al., 2023; Macgregor, 2024). In order to apply the superior clustering performance of spectral clustering to large-scale data, many efficient spectral clustering algorithms have been proposed in recent years. In order to reduce the high complexity of similarity matrix construction, the main idea is to construct a sparse graph, which not only reduces the time and memory of graph construction but also speeds up the subsequent eigendecomposition process (Spielman and Srivastava, 2011; Zhang et al., 2018; He et al., 2020; Liu et al., 2022). Nyström approximation is a simple method to construct a sparse graph (Fowlkes et al., 2004; Musco and Musco, 2017; Yang et al., 2012; Chen and Cai, 2011), but the clustering performance is greatly affected by the sampling points, so the landmark method based on $k$-means is proposed as an improvement (Bouneffouf and Birol, 2015; Rafailidis et al., 2017; Huang et al., 2019; Li et al., 2022; Xie et al., 2023). These methods reduce the time of graph construction from $\mathcal{O}(n^2)$ to $\mathcal{O}(n)$. In order to reduce the time complexity of eigendecomposition, using transfer cut (Li et al., 2012; Huang et al., 2019; Li et al., 2022) on the constructed $n \times p$ sparse graph can achieve eigendecomposition with $\mathcal{O}(n)$ (Huang et al., 2019; Li et al., 2022) or $\mathcal{O}(p^3)$ ($p$ is the number of landmarks) (Xie et al., 2023) time complexity.

These methods achieve faster spectral decomposition by constructing a smaller graph to replace the original graph. However, whether the Nyström-based or the landmark-based method, some information of the graph will be lost, which could reduce clustering performance. *Is it possible to speed up spectral decomposition while almost not losing information of the graph?*

In order to reduce information loss of the original graph, the number ($p$) of landmarks cannot be too small. Since we need a k-dimensional feature vector to indicate the category of each point, we need to perform eigendecomposition on a $k \times k$ matrix at least, while existing methods require the number of landmarks $p \gg k$. *Is it possible to perform efficient spectral clustering by only performing eigendecomposition on a $k \times k$ matrix?*

Existing methods are powerless to answer these two questions because they are all based on point perspectives, the limited sampled points cannot effectively represent the original graph. **A distribution-based perspective is a feasible way to answer the above two questions affirmatively**. In this paper, we achieve graph sparsification by constructing a bipartite graph ($n \times k$) between each point and $k$ distributions, so we only need to perform eigendecomposition on the $k \times k$ matrix. Since the distribution representation of the graph is employed, almost no information of the graph is lost. Figure 1 shows an example graph $\mathcal{G}$ that contains 3 subgraphs. We show the eigenvalues (first row) and eigenvectors (second row) of the eigendecomposition of the original graph (graph $\mathcal{G}$), the landmark-based method (point-based $\mathcal{B}$ and graph $\mathcal{G}_R$), and the distribution-based graph (distribution-based $\mathcal{B}$ and graph $\mathcal{G}_\Phi$), as well as the Normalized Mutual Information (NMI) (McDaid et al., 2011) scores using $k$-means (Wu et al., 2008) in the $\mathbb{R}^k$ space.

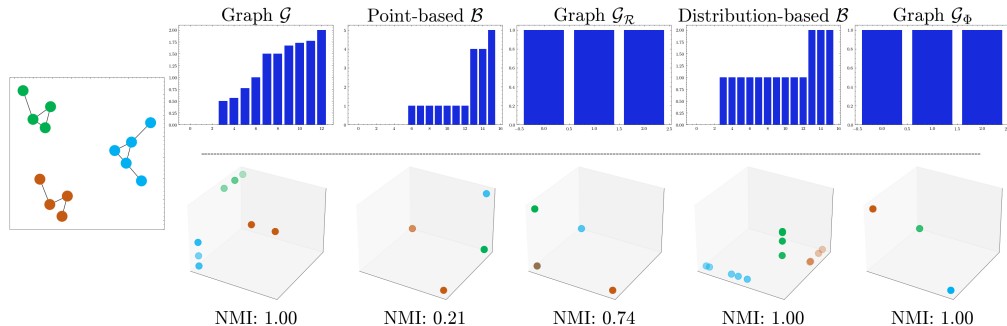

Figure 1: An example compares the distribution-based method with the point-based method. The distribution-based method contains more information about the original graph than the point-based method and achieves a better clustering effect. (Since graph $\mathcal{G}$ is composed of three subgraphs, the three eigenvalues of the normalized Laplacian matrix of graph $\mathcal{G}$, $\mathcal{B}$ are 0, and the three eigenvalues of the normalized adjacency matrix of graph $\mathcal{G}_R$, $\mathcal{G}_\Phi$ are 1.)

Using this idea, we propose a superior and effective distribution-based spectral clustering (D-SPEC) algorithm, We evaluate the proposed D-SPEC on a wide range of synthetic and real datasets ranging from 3 hundred to 20 million data points. The results demonstrate that the proposed D-SPEC algorithm affirmatively addresses the two aforementioned questions.

We summarize our contributions below:

- Enhancing the efficiency and effectiveness of spectral clustering by transitioning from a traditional point-based perspective to a distribution-based perspective.

- Proposing a distribution-based spectral clustering algorithm, termed D-SPEC, that only requires the eigendecomposition of a $k \times k$ matrix.

- Proving theoretically that D-SPEC retains the graph information and providing a bound for noise tolerance, indicate the enhanced robustness of D-SPEC.

- Demonstrating that our proposed D-SPEC outperforms existing methods through extensive experiments.

## 2    RELATED WORK

In this section, we provide a concise overview of the related work. An exhaustive related work have been included in the Appendix B to avoid excessive length. Spectral clustering aims to partition the data points into $k$ clusters using the spectrum of the graph Laplacian (Von Luxburg, 2007). Spectral clustering first constructs a similarity matrix between points. After calculating the Laplacian matrix, it

performs eigendecomposition to map the points to the $\mathbb{R}^k$ space. Finally, $k$-means is employed in the $\mathbb{R}^k$ space to complete the clustering. Although spectral clustering has good clustering performance and theoretical basis, its high time complexity limits its extension to large-scale data. In recent years, many algorithms have been developed to accelerate it.

Nyström approximation-based spectral clustering (Fowlkes et al., 2004; Musco and Musco, 2017; Yang et al., 2012; Chen and Cai, 2011) first randomly selects a small subset to construct a similarity sub-matrix, which can efficiently construct the similarity matrix and eigendecomposition, but the clustering performance is greatly affected by the sampling points. The landmark-based methods employ $k$-means to improve the cluster performance (Bouneffouf and Birol, 2015; Rafailidis et al., 2017; Li et al., 2012; Huang et al., 2019; Li et al., 2022; Xie et al., 2023).

In addition, some methods approximate the similarity of the original graph by random feature mapping, thereby efficiently constructing the similarity matrix and accelerating the eigendecomposition (Hansen and Mahoney, 2014; Wu et al., 2018; Rahman and Bouguila, 2020). Some methods calculate the approximate eigenvector by power method without eigendecomposition (Macgregor, 2024; Boutsidis and Magdon-Ismail, 2013).

## 3 ALGORITHM DESCRIPTION AND ANALYSIS

We now introduce our distribution-based spectral clustering[1] (D-SPEC), which constructs a bipartite graph between $n$ points and $k$ distributions, thus only requiring eigendecomposition of a $k \times k$ matrix. We also analyze the properties of D-SPEC in terms of preserving graph information and being robust to noise.

### 3.1 DISTRIBUTION-BASED SPECTRAL CLUSTERING

Given a dataset $\mathbb{X} \in \mathbb{R}^d = \{x_1, \ldots, x_n\}$, Spectral clustering (SC) constructs a fully connected undirected graph $\mathcal{G} = \{\mathbb{X}, \boldsymbol{W}\}$ with affinity matrix $\boldsymbol{W}$, where the element $w_{ij}$ of $\boldsymbol{W}$ indicates the similarity between points $x_i$ and $x_j$. The Laplacian matrix of $\mathcal{G}$ is $\boldsymbol{L} = \boldsymbol{D} - \boldsymbol{W}$, where $\boldsymbol{D}$ is the diagonal matrix with element $d_{ii} = \sum_{j \neq i} w_{ij}$. After eigendecomposition of the normalized Laplacian $\boldsymbol{N} = \boldsymbol{D}^{-\frac{1}{2}} \boldsymbol{L} \boldsymbol{D}^{\frac{1}{2}}$, the eigenvectors $\boldsymbol{u}_1, \ldots, \boldsymbol{u}_k$ corresponding to the smallest $k$ eigenvalues $\lambda_1, \ldots, \lambda_k$ are taken as the new feature $\mathbb{X}_{spec} \in \mathbb{R}^k$, Finally, clustering is completed using $k$-means on $\mathbb{X}_{spec} \in \mathbb{R}^k$.

Instead of a $n \times n$ fully connected graph, we construct an $n \times k$ bipartite graph $\mathcal{B} = \{\mathbb{X}, \mathbb{K}, \boldsymbol{W}^\Phi\}$ with similarity $\boldsymbol{W}^\Phi$ ($n \times k$), where the element $w_{ij}^\Phi$ represents the similarity between node $x_i$ and subgraph $\mathcal{G}_j$ of $\mathcal{G}$, which is measured by the similarity between point $x_i$ and distribution $P(\mathbb{C}_j)$, where $\mathbb{C}_j = \{x_1^j, \ldots, x_{nk}^j\}$ ($\bigcup_j \mathbb{C}_j = \mathbb{X}$) indicates the $j$-$th$ cluster. The $P(\mathbb{C}_j)$ is obtained in two steps: i) acquire cluster $\mathbb{C}_j$, ii) compute $P(\mathbb{C}_j)$ based on $\mathbb{C}_j$.

**Assumption 3.1.** Let $\mathcal{G} = \{\mathbb{X}, \boldsymbol{W}\}$ be the fully connected undirected graph formed by $\mathbb{X}$ with affinity matrix $\boldsymbol{W}$, then $\hat{w}(x, y) > \hat{w}(x, z).\forall x \in \mathbb{C}_i, z \in \mathbb{C}_j, j \neq i, i = \{1, \ldots, k\}$. Where $\hat{w}(x, y)$ is the similarity between point $x \in \mathbb{C}_i$ and its nearest neighbor $y \in \mathbb{C}_i$, and $\hat{w}(x, z)$ is the similarity between point $x$ and $z$ in other $\mathbb{C}_j (j \neq i)$.

*Acquiring cluster* $\mathbb{C}_j$: D-SPEC first maps the points into the Reproducing Kernel Hilbert Space (RKHS) $\mathbb{H}$ and constructs a fully connected undirected graph $\mathcal{G}_\mathbb{H}$ in the space $\mathbb{H}$ with affinity matrix $\boldsymbol{S}$, where $s_{ij}$ is the similarity between point $x_i$ and point $x_j$ in the space $\mathbb{H}$. According to Assumption 3.1, there exists a threshold $\tau$ to construct a bounded graph $\mathcal{G}_b$, wherein edges between clusters are removed, and only edges within the same cluster are maintained as shown in Figure 2. The nodes set $\mathbb{G}_j$ of the largest $k$ subgraphs $\mathcal{G}_j^b = \{\mathbb{G}_j, \boldsymbol{W}_j\}$ are selected as the approximation of $\mathbb{C}_j$. However, Assumption 3.1 is not universally satisfied for all points, as real-world datasets typically contain noise points. In order to eliminate the influence of noise, D-SPEC first samples $p$ points to construct graph $\mathcal{G}_s$ (instead of $\mathcal{G}_\mathbb{H}$) that meet Assumption 3.1.

---

[1]The codes and datasets are available at `https://anonymous.4open.science/r/D-SPEC/`.

*Computing $P(\mathbb{C}_j)$:* After obtaining the approximate cluster $\mathbb{G}_j$ of $\mathbb{C}_j$, D-SPEC employs kernel mean embedding to estimate $\mathcal{G}_j$:

$$\hat{\Phi}(\mathcal{G}_j) \approx \hat{\Phi}(\mathcal{G}_j^b) = \frac{\sum_{x \in \mathcal{G}_j^b} \Phi(x)}{|\mathcal{G}_j^b|} = \frac{\sum_{x \in \mathbb{G}_j} \Phi(x)}{|\mathbb{G}_j|} \tag{1}$$

where $\Phi(x)$ is the feature map of point $x$ in RKHS. The Gaussian kernel cannot be used to calculate Equation 1 due to its infinite-dimensional feature map, so we use the recently proposed Isolation Distribution Kernel (IDK) with finite dimensions (Ting et al., 2021). The similarity $w_{ij}^{\Phi}$ between node $x_i$ and subgraph $\mathcal{G}_j$ is the inner product of their feature maps: $w_{ij}^{\Phi} = \left\langle \Phi(x_i), \hat{\Phi}(\mathcal{G}_j) \right\rangle$.

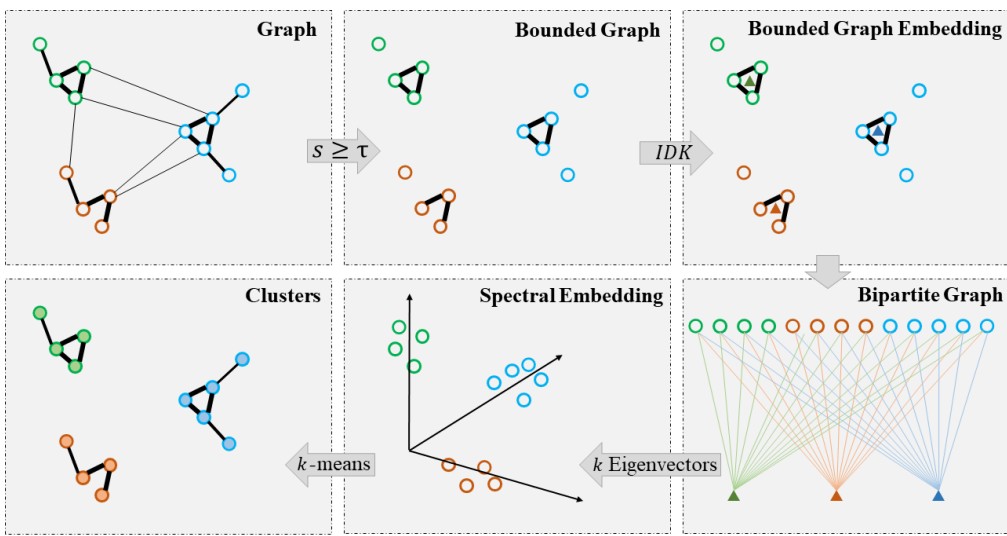

Figure 2: Illustration of D-SPEC. A Bounded graph is obtained by the threshold $\tau$, and then the distribution of each subgraph is obtained using Equation 1. A bipartite graph of nodes and distributions is constructed, and then eigendecomposition is performed to map the data to $\mathbb{R}^k$. Finally, the $k$-means clustering is employed in $\mathbb{R}^k$.

---

**Algorithm 1** D-SPEC: Distribution-based Spectral Clustering

---

**Require:** Graph $\mathcal{G} = \{\mathbb{X}, \boldsymbol{W}\}$, number of clusters $k$, threshold $\tau$
**Ensure:** Clustering result
  1: Sample subgraph $\mathcal{G}_s$ from $\mathcal{G}$
  2: Obtain bounded graph $\mathcal{G}_b = \{\mathcal{G}_1^b, \ldots, \mathcal{G}_k^b\}$
  3: Compute mapping $\hat{\Phi}(\mathcal{G}_j)$ using Equation 1
  4: Construct bipartite graph $\mathcal{B}$ between $\mathbb{X}$ and $\hat{\Phi}(\mathcal{G}_j)$
  5: Perform transfer cut to obtain $k$-dimensional spectral embedding
  6: Apply $k$-means clustering to the spectral embedding
  7: **return** Clustering result

---

After constructing the bipartite graph $\mathcal{B}$, if we regard $\mathcal{B}$ as a normal graph containing $n + k$ nodes, its affinity matrix [2] is:

$$\boldsymbol{B} = \begin{bmatrix} 0 & \boldsymbol{W}^{\top} \\ \boldsymbol{W} & 0 \end{bmatrix}$$

The time complexity of solving the eigen-problem $\boldsymbol{L}u = \gamma \boldsymbol{D}u$ is $\mathcal{O}(N + k)^3$, where $\boldsymbol{L} = \boldsymbol{D} - \boldsymbol{B}$. It is not computationally feasible for very large-scale datasets. Fortunately, we can employ the transfer cut (Li et al., 2012; Huang et al., 2019; Li et al., 2022) method to alleviate this complexity in the

---

[2]For notational convenience, let $\boldsymbol{W}$ denote $\boldsymbol{W}^{\Phi}$ henceforth.

bipartite graph. Let $\mathcal{G}_{\mathbb{K}}$ be the graph $\mathcal{G}_{\mathbb{K}} = \{\mathbb{K}, \boldsymbol{W}_{\mathbb{K}}\}$, where $\mathbb{K}$ is the node set, $\boldsymbol{W}_{\mathbb{K}} = \boldsymbol{W}^{\top}\boldsymbol{D}\boldsymbol{W}$ is the affinity matrix. Solving the new eigen-problem $\boldsymbol{L}_{\mathbb{K}}v = \lambda\boldsymbol{D}_{\mathbb{K}}v$ only requires eigendecomposition of the $k \times k$ matrix, which only demands the time complexity of $\mathcal{O}(nk^2)$ (Li et al., 2012; Huang et al., 2019; Li et al., 2022).

After applying the transfer cut to derive the spectral embedding comprising $k$ eigenvectors, $k$-means clustering can be subsequently utilized to accomplish the final clustering. Algorithm 1 shows the pseudo code of D-SPEC.

## 3.2 D-SPEC PRESERVES THE INFORMATION OF THE GRAPH AND IS ROBUST TO NOISE

**Theorem 3.2.** *Given a unweighted graph $\mathcal{G}$ that encompasses $k$ subgraphs, the D-SPEC yields $k$ zero eigenvalues. The matrix formed by the resulting eigenvectors has a single element of 1 in each row, indicating the cluster of each node, and all other elements are zero.*

Theorem 3.2 shows that D-SPEC does not lose graph information, while point-based methods cannot guarantee this, as shown in figure 1. For the case of noise, such as edge connections between different clusters (Appendix C), similar results can be obtained by using $\tau$ to select bounded graphs. If $\tau$ is not used to exclude these noises, distribution-based methods are also more robust.

**Theorem 3.3.** *Let $\mathcal{G}$ be a graph that does not consist of $k$ completely disjoint connected components, which means there are edges that connect vertices from different clusters. Let $\lambda_i^{(0)}$ be the $i$-th eigenvalue of the Laplacian matrix of the graph that does not contain edges between different clusters, $\lambda_i^{(1)}$ be the $i$-th eigenvalue of the Laplacian matrix of the graph $G$, and $\lambda_{di}^{(0)}$, $\lambda_{di}^{(1)}$ be the $i$-th eigenvalue of D-SPEC with and without noise respectively, then:*

$$\sup_{i\in\{1,...,n\}} |\lambda_{di}^{(1)} - \lambda_{di}^{(0)}| \leq \sup_{i\in\{1,...,n\}} |\lambda_i^{(1)} - \lambda_i^{(0)}|.$$

**Theorem 3.4.** *Let $d(V, V')$ denote the distance between the spectral embedding $V$ without noise and $V'$ with noise, and $d(V_d, V_d')$ denote the distance between the spectral embeddings obtained by D-SPEC, then:*

$$d(V, V') \leq \frac{C}{\lambda_{k+1}^{(0)}}, \ d(V_d, V_d') \leq \frac{C_d}{\lambda_{dk+1}^{(0)}}, \ \frac{C_d}{\lambda_{dk+1}^{(0)}} \leq \frac{C}{\lambda_{k+1}^{(0)}},$$

*where $C, C_d > 0$ are constant.*

Theorems 3.3 points out that the supremum of the difference between the eigenvalues of the graph with noise and the eigenvalues of the noise-free graph obtained by D-SPEC is smaller than that of using SC. That is to say, the eigenvalues obtained by using D-SPEC are closer to the eigenvalues of the noise-free graph. Theorem 3.4 shows that using D-SPEC can obtain spectral embeddings that are closer to the noise-free graph. Therefore, DSPEC can obtain more robust spectral embeddings and thus achieve more robust clustering.

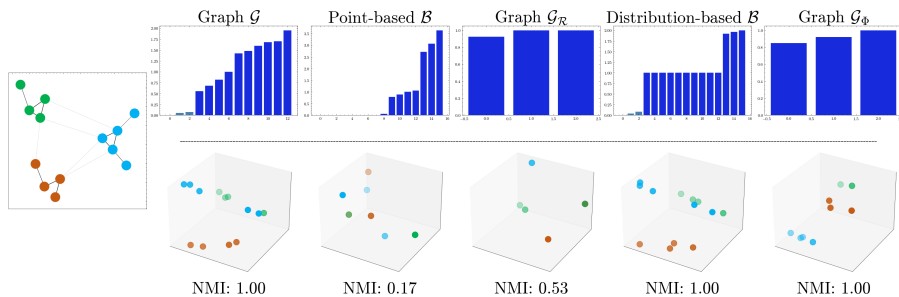

Figure 3: An example with noisy edges. (We do eigendecomposition on the normalized Laplacian matrix of graph $\mathcal{G}$, $\mathcal{B}$, and do eigendecomposition on the normalized adjacency matrix of graph $\mathcal{G}_R$, $\mathcal{G}_\Phi$.

An example with noise is given in Figure 3, the bipartite graph constructed by distribution is more robust than constructed by points.

## 4 EXPERIMENTS

In this section, we empirically study whether eigendecomposition of only $k \times k$ matrix can achieve effective clustering. We demonstrate the superiority of D-SPEC through the following four comparisons: (a) Performance on the benchmark datasets. (b) Scalability on large-scale datasets. (c) Performance on the fundamental limitations of spectral clustering. (d) A comparison of the ensemble version of D-SPEC and U-SPEC.

We compare the proposed D-SPEC with: SC (Shi and Malik, 2000): The original spectral clustering (Ncut) is used as our baseline. U-SPEC (Huang et al., 2019): Ultra-Scalable Spectral Clustering, A hybrid landmark selection method that combines random initialization of candidate samples with $k$-means to determine cluster centroids as representatives, followed by the computation of approximate K-nearest neighbor representatives. DNCSC (Li et al., 2022): Divide-and-conquer spectral clustering, a divide-and-conquer based landmark selection method to generate high-quality landmarks. FastSC (Macgregor, 2024): Fast spectral clustering method using power method to calculate approximate eigenvectors. GBSC (Xie et al., 2023): Spectral clustering algorithm based on granular-ball, which generates $p$ granular-balls from the original data, and performs spectral clustering only on the $p$ granular-balls.

We use Normalized Mutual Information (NMI) (McDaid et al., 2011), Adjusted Rand Index (ARI) (Rand, 1971; Gates and Ahn, 2017) and F-measure (Van Rijsbergen, 1977) as evaluation metrics. The experiments are executed on a Linux machine with 1T GB RAM and an AMD 128-core CPU, with each core running at 2 GHz.

### 4.1 EXPERIMENTS ON BENCHMARK DATASETS

We use fifteen datasets including the datasets used in the U-SPEC and DNSCS papers. The results in terms of NMI are shown in Table 1, The results in terms of ARI and F-measure are shown in Appendix D. SC requires a significant amount of time for clustering, exceeding two days for datasets larger than MNIST in size. GBSC is not able to handle the datasets large than mnist due to the memory consumption. Our D-SPEC method achieves the best scores on most of the fifteen benchmark datasets.

Table 1: Average NMI scores over 10 runs. The best score in each dataset is highlighted in bold.

| dataset | n | d | k | SC | U-SPEC | DNCSC | FastSC | GBSC | D-SPEC |
|---|---|---|---|---|---|---|---|---|---|
| spiral | 312 | 2 | 3 | **1.000** | **1.000** | **1.000** | 0.693 | 0.009 | **1.000** |
| 4C | 1000 | 2 | 4 | **1.000** | **1.000** | **1.000** | 0.726 | 0.528 | **1.000** |
| AC | 1004 | 2 | 2 | **1.000** | **1.000** | **1.000** | 0.340 | 0.610 | **1.000** |
| RingG | 1536 | 2 | 4 | 0.794 | 0.845 | 0.761 | 0.779 | 0.694 | **0.987** |
| complex9 | 3031 | 2 | 9 | **1.000** | 0.971 | 0.951 | 0.810 | 0.662 | **1.000** |
| cure-t2-4k | 4200 | 2 | 7 | 0.843 | 0.886 | 0.872 | 0.810 | 0.772 | **0.951** |
| landsat | 2000 | 36 | 6 | 0.281 | 0.668 | 0.647 | **0.740** | 0.646 | 0.647 |
| spambase | 4601 | 57 | 2 | 0.011 | 0.013 | 0.033 | 0.020 | 0.162 | **0.166** |
| waveform3 | 5000 | 21 | 3 | 0.371 | 0.370 | 0.369 | **0.605** | 0.370 | 0.406 |
| pendigits | 10992 | 16 | 10 | 0.641 | 0.826 | 0.813 | 0.523 | 0.596 | **0.847** |
| usps | 11000 | 256 | 10 | 0.676 | 0.654 | 0.652 | 0.564 | 0.338 | **0.778** |
| letters | 20000 | 16 | 26 | 0.278 | 0.455 | 0.437 | 0.308 | O/M | **0.478** |
| mnist | 70000 | 784 | 10 | **0.766** | 0.699 | 0.736 | 0.651 | O/M | 0.746 |
| skin | 245057 | 3 | 2 | N/A | 0.025 | 0.508 | 0.001 | O/M | **0.767** |
| covertype | 581012 | 54 | 7 | N/A | 0.212 | 0.086 | **0.695** | O/M | 0.218 |
| | Avg.score | | | 0.666 | 0.642 | 0.658 | 0.551 | 0.490 | **0.733** |
| | Avg.rank | | | 3.567 | 2.833 | 3.433 | 4.067 | 5.400 | **1.700** |

* O/M indicates the out-of-memory error.
* N/A indicates that no clustering results were obtained within two days.

The Nemenyi significance (Nemenyi, 1963) test results are shown in Figure 4. D-SPEC ourperforms the other methods in all three metrics, and only D-SPEC is significantly better than SC. Because D-SPEC retains the information of the graph through the distribution representation of the graph, and removes the noise information in the graph through the bounded graph.

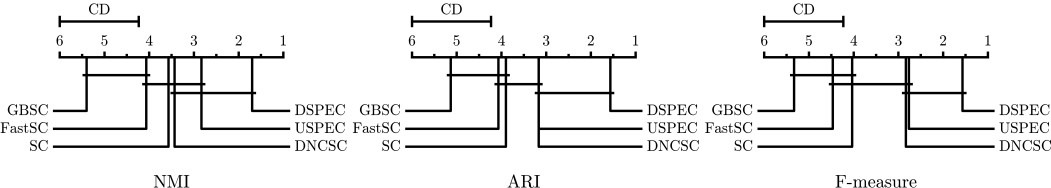

Figure 4: The Nemenyi significance test results at 0.1 significance level.

For the experimental verification of Theorems 3.3 and 3.4 that our method is more robust to noise. We added uniform noise $U(0,1)$ with different ratios (0-0.2) to the smallest 4C (spiral is too small) and the largest usps from the dataset where all algorithms can run. The results are shown in Figure 5.

On 4C, all algorithms generally show a trend of worsening performance as the ratio increases. When the noise ratio reaches 0.2, only DNCSC and D-SPEC achieve an NMI greater than 0.9, while the NMI of SC is 0.52. On usps, only U-SPEC, DNCSC and D-SPEC are almost unaffected by noise. This may be because the clustering results of U-SPEC and DNCSC are not very high in the absence of noise. They are all around 0.65, much lower than 0.78 of D-SPEC. For this reason, we compare them on the second largest dataset pendigits, because they all perform well (¿0.8) and have closed results when there is no noise in this data. As the ratio of noise increases, the NMI of U-SPEC decreases from 0.83 to 0.79, the NMI of DNCSC decreases from 0.81 to 0.76, and only the NMI of D-SPEC remains almost unchanged.

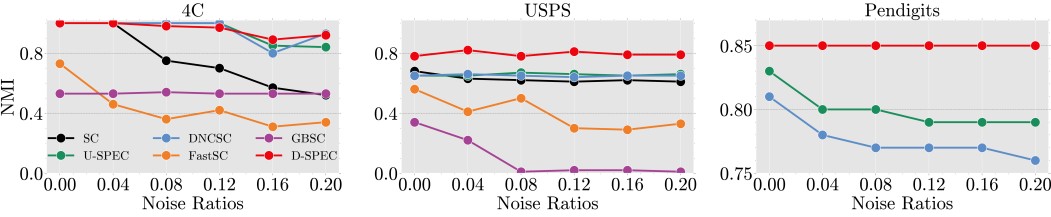

Figure 5: Comparison of robustness to noise.

## 4.2 COMPARISON ON LARGE-SCALE DATASETS

In this subsection, we summarize the time complexity of D-SPEC, and compare D-SPEC with other algorithms on large-scale datasets.

To obtain $k$ distributions, D-SPEC first samples $p$ points from $\mathbb{X}$ to construct $\mathcal{G}_s$, and then obtains the distribution of each subgraph according to Equation 1 on the bounded graph, which takes $\mathcal{O}(p^2)$ time. Constructing a bipartite graph is to calculate the similarity between $n$ points and $k$ distributions, which takes $\mathcal{O}(nk)$ time. Finally, transfer cut takes $\mathcal{O}(nk^2 + k^3)$ time, includes the eigendecomposition time of $\mathcal{O}(k^3)$. Table 2 provides a comparison of computational complexity of our D-SPEC algorithm against other large-scale spectral clustering algorithms.

In order to demonstrate that D-SPEC can achieve efficient and effective clustering for large-scale data by only performing eigendecomposition on a $k \times k$ matrix, we selected the five large-scale datasets used in U-SPEC, with points ranging from 1 million to 20 million. We compared the NMI and runtime of these algorithms on these large-scale data, as shown in the Figure 6, and the results of ARI and F-measure are in the Appendix I.

The gray bar in the figure indicates that the algorithm cannot handle the dataset. SC and GBSC failed on all five datasets. FastSC can run on the first two datasets, but the clustering effect is very poor because it uses approximate eigenvectors. DNCSC overflowed the memory on the largest dataset.

Table 2: Comparison of the computational complexity.

|  | SC | USPEC | DNCSC | FastSC | GBSC | D-SPEC |
|---|---|---|---|---|---|---|
| LS | N/A | $\mathcal{O}(p^2)$ | $\mathcal{O}(n\alpha)$ | N/A | $\mathcal{O}(n\log n)$ | $\mathcal{O}(p^2)$ |
| Sco | $\mathcal{O}(n^2)$ | $\mathcal{O}(np^{\frac{1}{2}})$ | $\mathcal{O}(nK)$ | $\mathcal{O}(n^2)$ | $\mathcal{O}(p^2)$ | $\mathcal{O}(nk)$ |
| ED | $\mathcal{O}(n^3)$ | $\mathcal{O}(nK(K+k)+p^3)$ | $\mathcal{O}(nK(K+k)+p^3)$ | $\mathcal{O}(\frac{nK}{\epsilon^2})$ | $\mathcal{O}(p^3)$ | $\mathcal{O}(nk^2+k^3)$ |

* LS: Landmark selection. Sco: Similarity construction. ED: Eigendecomposition.
* n: number of points. k :number of clusters. p: number of landmards. K: number of nearest neighbors.
* N/A indicates that no such step in the algorithm.

Only USPEC and D-SPEC can handle all datasets, and D-SPEC achieves the best clustering results on all datasets.

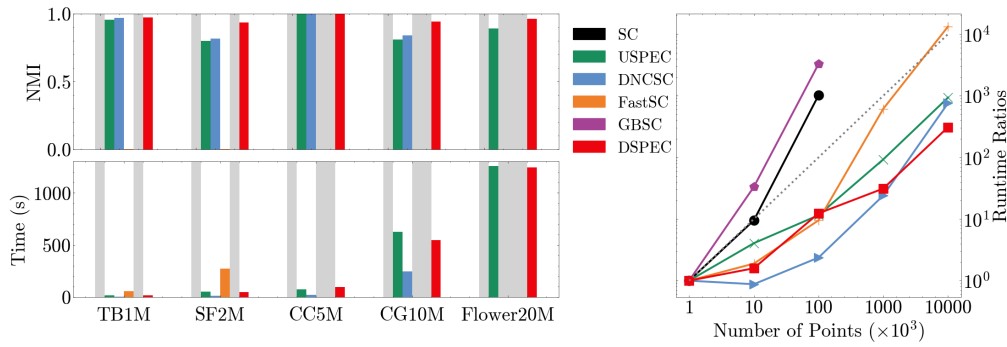

Figure 6: Results in terms of NMI and runtime in 5 large-scale datasets (left). And scale-up test on CG10M dataset (right).

Since the running time of an algorithm is not only related to the number of points in the dataset, but also to the distribution, dimension and the number of clusters of the data, in order to compare the running time of D-SPEC and other algorithms that only change with the size of the dataset, we randomly sampled 1k, 10k, 100k, 1M, and 10M points from CG-10M dataset, and each algorithm was experimented with the same parameters on all datasets. The results are shown in the Figure 6 (right), where the vertical axis (logarithmized) is the ratio of the running time on different datasets to the running time on the 1k dataset. The results are consistent with Table 2. D-SPEC has a very low time complexity, especially on 1M and 10M dataset.

In short, experiments on large-scale datasets show that the D-SPEC algorithm, which only performs eigendecomposition on the $k \times k$ matrix, is effective and efficient.

### 4.3 FUNDAMENTAL LIMITATIONS OF SPECTRAL CLUSTERING

Nadler and Galun (2006) pointed out that spectral clustering has some fundamental limitations. We compare it with the three example datasets given by (Nadler and Galun, 2006). The experimental results of these three datasets are shown in Figure 7.

D-SPEC is the only algorithm that maintains good performance on all three datasets. There are two main reasons. (a) D-SPEC first extracts a bounded graph, which removes some noise edges, such as those caused by cluster overlap. (b) D-SPEC constructs a bipartite graph based on distribution, which can preserve cluster information.

For example, the two clusters in the second dataset are not ignored (which would be ignored by other clustering algorithms because each cluster has very few points), and the uniform distribution line cluster in the third dataset span a large area, but their distribution information is preserved, thus achieving more effective spectral clustering. In a nutshell, D-SPEC provides a possible way to overcome the basic limitations of spectral clustering.

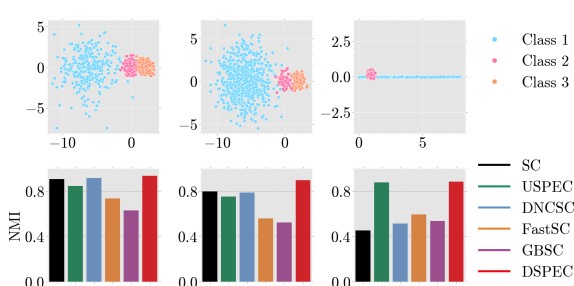
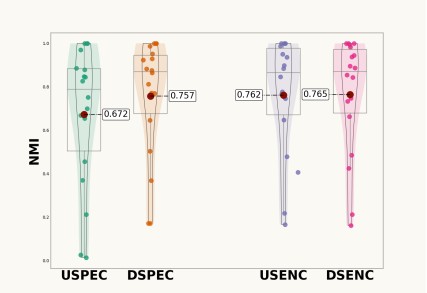

Figure 7: Comparison of algorithms for addressing the fundamental limitations of spectral clustering.

Figure 8: Comparison between U-SPEC and D-SPEC and their ensemble versions.

The first row of figure illustrates the dataset, while the second row presents the NMI scores of various clustering algorithms. In the first dataset, which is composed of three clusters with an equal point ratio of 1:1:1 and a density ratio of 1:8:8, all algorithms perform well except for FastSC and GBSC. However, when the point ratio is altered to 8:1:1 (second dataset), the NMI scores of all algorithms except D-SPEC decline below 0.8. For the third dataset consisting of a Gaussian distribution and a uniform distribution, the NMI of all algorithms except U-SPEC and D-SPEC is poor and below 0.6.

### 4.4 Ensemble Distribution-based Spectral clustering

Ensemble learning is often used to combine multiple base model to improve the performance of the base model. For example, the U-SENC (Huang et al., 2019), proposed as the ensemble version of U-SPEC, significantly improved the clustering quality of the U-SPEC. We compared D-SPEC with U-SENC, the results are shown in Figure 8. Even compared with U-SENC, the overall performance of our proposed D-SPEC algorithm is better.

After we performed the same ensemble method on D-SPEC, we found that the improvement in clustering performance was very small (as shown in the Figure 8). The reason is that ensemble learning often works on weak base models, while D-SPEC has good clustering performance. In addition, ensemble learning requires a large diversity. Since U-SPEC is point-based, the diversity is large, while the diversity based on distribution is small.

Therefore, the ensemble methods currently used in spectral clustering cannot improve the performance of D-SPEC. How to increase the diversity of D-SPEC so that it can achieve better performance using ensemble learning is an open question.

U-SENC first employs U-SPEC to cluster the data into $k'$ clusters, where $k' \in \{k \in \mathbb{Z} | 20 \le k \le 60\}$, and repeats $m$ times. Then, based on the clustering results, a bipartite graph of $n \times \hat{k}$ is constructed, where $\hat{k} = \sum_{i=1}^{m} k'_i$. Finally, spectral clustering is used on the bipartite graph to obtain the final clustering result. We employed the same ensemble method on D-SPEC termed D-SENC (DSENC in the figure). The numerical values in the figure represent the average scores in terms of NMI on the datasets.

## 5 Conclusion

In this paper, we propose an efficient distribution-based spectral clustering algorithm. The algorithm constructs a bipartite graph of $n$ graph nodes and $k$ distributions, thus achieving a affirmative answer to the two questions at the beginning of this paper: 1) fast spectral clustering is achieved with almost no loss of graph information. 2) effective spectral clustering can be achieved by only performing eigendecomposition on a $k \times k$ matrix. We theoretically prove that distribution-based spectral clustering can preserve graph information and is more robust to noise. We experimentally show that our proposed D-SPEC has better clustering performance than existing fast spectral clustering algorithms, and provides a way to address the fundamental limitations of spectral clustering.

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

## A   LARGE LANGUAGE MODELS

We used Large Language Models to polish our writing.

## B   EXTENDED REVIEW OF RELATED WORK

Spectral clustering has emerged as a powerful technique for partitioning data based on the eigenvectors of similarity matrices. Its ability to identify complex cluster structures makes it widely applicable across various domains, including image segmentation, bioinformatics, and social network analysis. However, the computational complexity associated with spectral clustering, particularly the eigendecomposition of large similarity matrices, poses significant challenges when dealing with large-scale datasets. To address these challenges, numerous methods have been developed to accelerate spectral clustering while maintaining its effectiveness.

Matrix decomposition and random feature mapping techniques have been extensively explored to reduce the computational burden of spectral clustering. The Nyström method (Fowlkes et al., 2004; Musco and Musco, 2017; Li et al., 2019; Yang et al., 2012) approximates large similarity matrices by sampling a subset of data points and performing eigendecomposition on a smaller submatrix, thereby decreasing the dimensionality of the problem. Rahimi and Recht (2007); Hansen and Mahoney (2014); Wu et al. (2018); Rahman and Bouguila (2020) introduced random feature mappings as an alternative approach to approximate kernel functions, facilitating efficient computation of similarity matrices in high-dimensional spaces. Building on these foundations, Xie et al. (2023) proposed an efficient spectral clustering algorithm based on granular-ball methods, which leverages random feature mappings to enhance scalability and precision.

Sampling-based approaches aim to reduce the computational load by performing spectral clustering on a representative subset of the data. Landmark-based methods are prominent approaches within this category. U-SPEC (Bouneffouf and Birol, 2015; Rafailidis et al., 2017; Li et al., 2012; Huang et al., 2019) introduces an ultra-scalable spectral clustering technique that constructs a bipartite graph between data points and landmarks. It employs the TransferCut algorithm to accelerate clustering, though the initial performance may be suboptimal. To address this, U-SENC integrates ensemble strategies, enhancing both efficiency and clustering quality. Additionally, landmark-based methods such as those proposed by Chen and Cai (2011) focus on selecting representative "landmark" points to anchor the clustering process, thereby reducing computational complexity. Furthermore, Li et al. (2022) presented divide-and-conquer strategies that partition data into subsets, perform local spectral clustering, and merge the results, effectively reducing computational complexity and enhancing scalability. These sampling-based methods ensure that spectral clustering remains feasible even for extremely large datasets by focusing computational efforts on strategically chosen subsets of data.

Beyond the above strategies, various other methods have been proposed to accelerate spectral clustering. Macgregor (2024); Boutsidis and Magdon-Ismail (2013) developed a fast and simple spectral clustering approach that demonstrates both theoretical and practical advantages over traditional methods by employing power method for accelerating spectral clustering. Additionally, graph sparsification methods (Spielman and Srivastava, 2011) aim to reduce the number of edges in the similarity graph while preserving its essential spectral properties, thereby enabling more efficient computations. Approximate spectral clustering algorithms (Tremblay et al., 2016; Ye et al., 2018; Wang et al., 2020) also contribute to scalability by employing techniques such as matrix sketching, compressive sensing, and the use of anchor graphs to achieve faster computations while maintaining clustering quality. These diverse approaches highlight the multifaceted efforts to tackle the scalability issues inherent in spectral clustering, each bringing unique strengths to the table.

In summary, existing advancements in accelerating spectral clustering can be broadly categorized into Nyström and random feature mapping methods, sampling-based approaches, and other innovative acceleration techniques. While these methods have significantly enhanced the scalability and efficiency of spectral clustering, they often involve trade-offs between computational speed and clustering quality.

## C   PROOF OF THE THEOREMS

**Theorem 3.2.** *Given a unweighted graph $\mathcal{G}$ that encompasses $k$ subgraphs, the D-SPEC yields $k$ zero eigenvalues. The matrix formed by the resulting eigenvectors has a single element of 1 in each row, indicating the category of each node, and all other elements are zero.*

*Proof.* The degree matrix $\boldsymbol{D}_{\mathbb{K}}$ of $\boldsymbol{W}_{\mathbb{K}}$ is a diagonal matrix with its $(i, i)$-th entry being the sum of the $i$-th row of $\boldsymbol{W}_{\mathbb{K}}$:

$$\boldsymbol{D}_{\mathbb{K}}(i, i) = \sum_{j} \boldsymbol{W}_{\mathbb{K}}(i, j). \boldsymbol{L} = \boldsymbol{D}_{\mathbb{K}} - \boldsymbol{W}_{\mathbb{K}}.$$

We construct an eigenvector:

$$v = [0, 0, \ldots, 1, \ldots, 0]^{\top}$$

In which only the $j$-th element is 1, and the rest are 0.

$$\boldsymbol{L}v_j = \boldsymbol{D}_{\mathbb{K}}v_j - \boldsymbol{W}_{\mathbb{K}}v_j$$

For $\boldsymbol{D}_{\mathbb{K}}v_i$, the result is the sum of the $j$-th column, denoted as $\boldsymbol{D}_{\mathbb{K}jj}$.
For $\boldsymbol{W}_{\mathbb{K}}v_j$, the result is $\boldsymbol{W}_{\mathbb{K}jj}$, which is the sum of the $j$-th subgraph.
So:

$$\boldsymbol{L}v_j = \boldsymbol{D}_{\mathbb{K}jj} - \boldsymbol{W}_{\mathbb{K}jj}.$$

Due to

$$\boldsymbol{D}_{\mathbb{K}jj} = \boldsymbol{W}_{\mathbb{K}jj}.$$

so:

$$\boldsymbol{L}v_j = 0.$$

This shows that $v_j$ is the eigenvector of $\boldsymbol{L}$ with eigenvalue 0. Since there are $k$ subgraphs, we can construct $k$ such eigenvectors $v_1, v_2, \ldots, v_k$, corresponding to $k$ 0 eigenvalues. □

**Lemma C.1.** *Given the eigenvector $\lambda$ of normalized Laplacian matrix and eigenvector $\gamma$ of normalized affinity matrix, then $\lambda + \gamma = 1$.*

*Proof.* Let

$$\boldsymbol{W}_n = \boldsymbol{D}^{-1/2}\boldsymbol{W}\boldsymbol{D}^{-1/2}.$$

be the normalised affinity matrix. and

$$\boldsymbol{L}_n = \boldsymbol{D}^{-1/2}(\boldsymbol{D} - \boldsymbol{W})\boldsymbol{D}^{-1/2} = \boldsymbol{I} - \boldsymbol{W}_n.$$

so $\lambda + \gamma = 1$. □

Let G = (V, E) be an undirected graph that does not consist of k strictly disjoint connected components,

**Theorem 3.3.** *Let $\mathcal{G}$ be a graph that does not consist of $k$ completely disjoint connected components, which means there are edges that connect vertices from different clusters. Let $\lambda_i^{(0)}$ be the $i$-th eigenvalue of the Laplacian matrix of the graph that does not contain edges between different clusters, $\lambda_i^{(1)}$ be the $i$-th eigenvalue of the Laplacian matrix of the graph $G$, and $\lambda_{di}^{(0)}$, $\lambda_{di}^{(1)}$ be the $i$-th eigenvalue of D-SPEC with and without noise respectively, then:*

$$\sup_{i\in\{1,\ldots,n\}} |\lambda_{di}^{(1)} - \lambda_{di}^{(0)}| \leq \sup_{i\in\{1,\ldots,n\}} |\lambda_i^{(1)} - \lambda_i^{(0)}|.$$

*Proof.* Let:
$\boldsymbol{L}$ be the Laplacian matrix of the graph $\mathcal{G}$ without noise (consisting of $k$ non-intersecting subgraphs),
$\boldsymbol{L}'$ be the Laplacian matrix of the graph $\mathcal{G}'$ with noise (there are edges between different subgraphs),
$\boldsymbol{L}_d$ be the Laplacian matrix of the graph $\mathcal{G}$ obtained by D-SPEC,
$\boldsymbol{L}_d'$ be the Laplacian matrix of the graph $\mathcal{G}'$ obtained by D-SPEC.

$$\Delta = \boldsymbol{L}' - \boldsymbol{L}, \Delta_d = \boldsymbol{L}_d' - \boldsymbol{L}_d.$$

According to the Wey inequality (Weyl, 1912), for $\boldsymbol{L}' = \boldsymbol{L} + \Delta$, we have

$$\lambda_k(\boldsymbol{L} + \Delta) \geq \lambda_k(\boldsymbol{L}) + \lambda_n(\Delta), \lambda_k(\boldsymbol{L}') \geq \lambda_k(\boldsymbol{L}) + \lambda_n(\Delta), k \in \{1,\ldots,n\}.$$

And,
$$\lambda_k(\boldsymbol{L}') \leq \lambda_k(\boldsymbol{L}) + \lambda_1(\Delta).$$

So:
$$\lambda_k(\boldsymbol{L}) + \lambda_n(\Delta) \leq \lambda_k(\boldsymbol{L}') \leq \lambda_k(\boldsymbol{L}) + \lambda_1(\Delta).$$
$$|\lambda_k(\boldsymbol{L}') - \lambda_k(\boldsymbol{L})| \leq \max\{|\lambda_1(\Delta)|, |\lambda_n(\Delta)|\}.$$

So,
$$|\lambda_k(\boldsymbol{L}') - \lambda_k(\boldsymbol{L})| \leq \sigma_1(\Delta).$$
$$|\lambda_k(\boldsymbol{L}_d') - \lambda_k(\boldsymbol{L}_d)| \leq \sigma_1(\Delta_d).$$

where $\sigma_1(\Delta)$ is the largest singular value of matrix $\Delta$.

The Rayleigh quotient of $\Delta$:
$$R(x) = \frac{x^\top \Delta^\top \Delta x}{x^\top x}.$$

Then
$$\sigma_1^2(\Delta) = \max_{x\neq 0} R(x).$$

The Rayleigh quotient of $\Delta$:
$$R_d(y) = \frac{y^\top \Delta_d^\top \Delta_d y}{y^\top y}.$$

Since $\Delta_d$ is a sub-matrix of $\Delta$,
$$y^\top \Delta_d^\top \Delta_d y \leq x^\top \Delta^\top \Delta x.$$

Then
$$\max_{y\neq 0} R_d(y) \leq \max_{x\neq 0} R(x),$$

thus
$$\sigma_1(\Delta_d) \leq \sigma_1(\Delta).$$

Hence,
$$\sup_{i\in\{1,\ldots,n\}} |\lambda_{di}^{(1)} - \lambda_{di}^{(0)}| \leq \sup_{i\in\{1,\ldots,n\}} |\lambda_i^{(1)} - \lambda_i^{(0)}|.$$

$\square$

**Theorem 3.4.** *Let $d(V, V')$ denote the distance between the spectral embedding $V$ without noise and $V'$ with noise, and $d(V_d, V'_d)$ denote the distance between the spectral embeddings obtained by D-SPEC, then:*

$$d(V, V') \leq \frac{C}{\lambda_{k+1}^{(0)}}, d(V_d, V'_d) \leq \frac{C_d}{\lambda_{dk+1}^{(0)}}, \frac{C_d}{\lambda_{dk+1}^{(0)}} \leq \frac{C}{\lambda_{k+1}^{(0)}},$$

*where $C, C_d > 0$ are constant.*

*Proof.* According to Davis-Kahan $\sin\Theta$ theorem, we have:

$$d(V, V') = \|\sin\Theta(V, V')\| \leq \frac{C}{\delta},$$

Specifically, we have:

$$d(V, V') \leq \frac{\|\Delta\|}{\lambda_{k+1}^0}.$$

The $k$-th eigenvalue represents the compactness of the graph. The graph constructed based on the distribution method is more compact, because

$$\langle \Phi(x), \Phi(\mathcal{G}_j) \rangle = \frac{\sum_{y \in \mathcal{G}_j} \langle \Phi(x), \Phi(y) \rangle}{|\mathcal{G}_j|}.$$

is equivalent to establishing edges between every two points in the subgraph $\mathcal{G}_j$. Therefore

$$\lambda_{dk+1}^{(0)} > \lambda_{k+1}^{(0)},$$

since $\Delta_d$ is a submatrix of $\Delta$,

$$\|\Delta_d\| < \|\Delta\|.$$

Therefore

$$\frac{C_d}{\lambda_{dk+1}^{(0)}} \leq \frac{C}{\lambda_{k+1}^{(0)}}.$$

$\square$

# D   MAIN RESULTS IN TERMS OF ARI AND F-MEASURE

The results in terms of ARI are shown in Table 3, And the results in terms of F-measure are shown in Table 3.

Table 3: Average ARI scores over 10 runs. The best score in each dataset is highlighted in bold.

| dataset | #n | #d | k | SC | U-SPEC | DNCSC | FastSC | GBSC | D-SPEC |
|---------|-----|-----|-----|-------|--------|-------|--------|------|--------|
| spiral | 312 | 2 | 3 | **1.000** | **1.000** | **1.000** | 0.612 | 0.004 | **1.000** |
| 4C | 1000 | 2 | 4 | **1.000** | **1.000** | **1.000** | 0.562 | 0.399 | **1.000** |
| AC | 1004 | 2 | 2 | **1.000** | **1.000** | **1.000** | 0.202 | 0.656 | **1.000** |
| RingG | 1536 | 2 | 3 | 0.571 | 0.776 | 0.522 | 0.662 | 0.582 | **0.987** |
| complex9 | 3031 | 2 | 9 | **1.000** | 0.932 | 0.896 | 0.691 | 0.391 | **1.000** |
| cure-t2-4k | 4200 | 2 | 7 | 0.889 | 0.920 | 0.869 | 0.707 | 0.620 | **0.951** |
| landsat | 2000 | 36 | 6 | 0.088 | 0.597 | 0.581 | 0.527 | 0.591 | **0.598** |
| spambase | 4601 | 57 | 2 | 0.005 | 0.001 | 0.019 | 0.029 | 0.160 | **0.250** |
| waveform3 | 5000 | 21 | 3 | 0.253 | 0.252 | 0.267 | **0.495** | 0.252 | 0.377 |
| pendigits | 10992 | 16 | 10 | 0.573 | 0.724 | 0.680 | 0.482 | 0.425 | **0.787** |
| usps | 11000 | 256 | 10 | 0.455 | 0.505 | 0.510 | 0.325 | 0.189 | **0.676** |
| letters | 20000 | 16 | 26 | 0.020 | 0.179 | 0.187 | 0.045 | O/M | **0.224** |
| mnist | 70000 | 784 | 10 | 0.604 | 0.611 | 0.658 | 0.464 | O/M | **0.677** |
| skin | 245057 | 3 | 2 | N/A | 0.001 | 0.565 | 0.020 | O/M | **0.878** |
| covertype | 581012 | 54 | 7 | N/A | 0.089 | 0.008 | **0.566** | O/M | 0.167 |
| Avg.score | | | | 0.574 | 0.572 | 0.584 | 0.426 | 0.388 | **0.705** |
| Avg.rank | | | | 3.900 | 3.167 | 3.167 | 4.067 | 5.133 | **1.567** |

Table 4: Average F-measure scores over 10 runs. The best score in each dataset is highlighted in bold.

| dataset | #n | #d | k | SC | U-SPEC | DNCSC | FastSC | GBSC | D-SPEC |
|---------|-----|-----|-----|-------|--------|-------|--------|------|--------|
| spiral | 312 | 2 | 3 | **1.000** | **1.000** | **1.000** | 0.667 | 0.394 | **1.000** |
| 4C | 1000 | 2 | 4 | **1.000** | **1.000** | **1.000** | 0.550 | 0.440 | **1.000** |
| AC | 1004 | 2 | 2 | **1.000** | **1.000** | **1.000** | 0.657 | 0.909 | **1.000** |
| RingG | 1536 | 2 | 3 | 0.666 | 0.728 | 0.604 | 0.636 | 0.633 | **0.987** |
| complex9 | 3031 | 2 | 9 | **1.000** | 0.939 | 0.903 | 0.587 | 0.522 | **1.000** |
| cure-t2-4k | 4200 | 2 | 7 | 0.772 | 0.811 | 0.832 | 0.579 | 0.759 | **0.941** |
| landsat | 2000 | 36 | 6 | 0.222 | 0.717 | 0.708 | 0.519 | 0.723 | **0.735** |
| spambase | 4601 | 57 | 2 | 0.374 | 0.391 | 0.428 | 0.377 | 0.320 | **0.721** |
| waveform3 | 5000 | 21 | 3 | 0.509 | 0.517 | 0.535 | 0.525 | 0.510 | **0.721** |
| pendigits | 10992 | 16 | 10 | 0.692 | 0.819 | 0.758 | 0.553 | 0.611 | **0.863** |
| usps | 11000 | 256 | 10 | 0.534 | 0.583 | 0.590 | 0.433 | 0.135 | **0.756** |
| letters | 20000 | 16 | 26 | 0.183 | 0.341 | 0.335 | 0.189 | O/M | **0.380** |
| mnist | 70000 | 784 | 10 | 0.623 | 0.711 | **0.750** | 0.539 | O/M | 0.747 |
| skin | 245057 | 3 | 2 | N/A | 0.442 | 0.808 | 0.519 | O/M | **0.953** |
| covertype | 581012 | 54 | 7 | N/A | 0.267 | 0.164 | **0.531** | O/M | 0.262 |
| Avg.score | | | | 0.660 | 0.684 | 0.694 | 0.524 | 0.541 | **0.804** |
| Avg.rank | | | | 4.033 | 2.767 | 2.833 | 4.467 | 5.333 | **1.567** |

# E LIMITATION OF D-SPEC

D-SPEC has two parameters, $\psi$ (parameter of IDK) and $\tau$ (parameter of D-SPEC). Since D-SPEC is a distribution-based method, its parameters are sensitive to the distribution of data. We show the impact of these two parameters on the Pendigits, and Mnist datasets, the results are shown in the Figure 9. Like most clustering algorithms, such as DBSCAN, DP, and Spectral clustering (the parameters of the Gaussian kernel), it is a limitation of D-SPEC that a suitable parameter must be selected to maximize the performance of the algorithm.

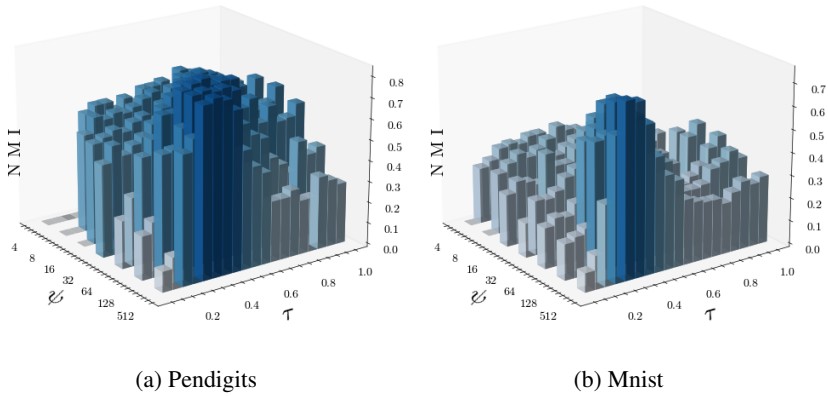

(a) Pendigits               (b) Mnist

Figure 9: The sensitivity analysis of parameters $\psi$ and $\tau$ of D-SPEC.

# F COMPARISON WITH DEEP SPECTRAL CLUSTERING

We compare with the recently deep spectral clustering DSC-LSP Meng et al. (2025), the results in terms of NMI are shown in Table 5. When we run DSC-LSP on the dataset we used, we get a very low NMI. We speculate this may be because we did not set the appropriate hyper-parameters (there are 7 hyper-parameters in DSC-LSP). Therefore, we compare it on the dataset used in the DSC-LSP paper and use the NMI reported in their paper. We find that DSC-LSP has better results for image data (such as mnist, fashion-mnist) because it can get better data representation, and spectral clustering based on this representation will be better, but for tabular data (such as mfeat-kar, isolet1234 and pendigits), DSC-LSP is not as good as D-SPEC.

Table 5: Results in terms of NMI scores compared with DSC-LSP.

| Dataset | #n | #d | k | D-SPEC | DSC-LSP |
|---|---|---|---|---|---|
| mnist | 70000 | 784 | 10 | 0.746 | **0.919** |
| mnist-test | 10000 | 784 | 10 | 0.773 | **0.784** |
| fashion-mnist | 70000 | 784 | 10 | 0.594 | **0.693** |
| cnae | 1080 | 856 | 9 | 0.597 | **0.599** |
| msra25-uni | 1799 | 256 | 12 | **0.713** | 0.704 |
| mfeat-kar | 2000 | 64 | 10 | **0.818** | 0.726 |
| isolet1234 | 6238 | 617 | 26 | **0.762** | 0.714 |
| pendigits | 10992 | 16 | 10 | **0.847** | 0.786 |

## G    COMPARISON WITH CORESET SPECTRAL CLUSTERING

Recently, a coreset-based spectral clustering algorithm CSC was proposed Jourdan et al. (2025). CSC first obtains the coreset in the kernel space, and then uses spectral clustering to cluster these coresets. After obtaining the $k$ centers of the coreset, all points are assigned to the nearest center to complete the clustering. CSC just replaces the step of using $k$-means to obtain $k$ centers in the coreset-based kernel k-means with using spectral clustering to obtain $k$ centers. which can be replaced by any other algorithm (such as GMM, etc.) In addition, using spectral clustering to replace $k$-means will increase the time complexity from $O(n)$ to $O(n^2)$. We compare the methods of D-SPEC and CSC as shown in Table 6. The coreset ratio was set to 0.1 on all datasets. D-SPEC outperforms CSC on all datasets used.

Table 6: Results compared with CSC.

| dataset | #n | #d | k | NMI | | ARI | | F1 | |
|---|---|---|---|---|---|---|---|---|---|
| | | | | D-SPEC | CSC | D-SPEC | CSC | D-SPEC | CSC |
| spiral | 312 | 2 | 3 | 1.000 | 0.156 | 1.000 | 0.055 | 1.000 | 0.488 |
| 4C | 1000 | 2 | 4 | 1.000 | 0.231 | 1.000 | 0.174 | 1.000 | 0.444 |
| AC | 1004 | 2 | 2 | 1.000 | 0.814 | 1.000 | 0.903 | 1.000 | 0.970 |
| RingG | 1536 | 2 | 3 | 0.987 | 0.596 | 0.987 | 0.569 | 0.987 | 0.607 |
| complex9 | 3031 | 2 | 9 | 1.000 | 0.683 | 0.999 | 0.416 | 0.999 | 0.437 |
| cure-t2-4k | 4200 | 2 | 7 | 0.951 | 0.801 | 0.951 | 0.721 | 0.941 | 0.775 |
| landsat | 2000 | 36 | 6 | 0.647 | 0.506 | 0.598 | 0.369 | 0.735 | 0.623 |
| spambase | 4601 | 57 | 2 | 0.166 | 0.040 | 0.250 | 0.077 | 0.721 | 0.607 |
| waveform3 | 5000 | 21 | 3 | 0.406 | 0.305 | 0.377 | 0.265 | 0.721 | 0.571 |
| pendigits | 10992 | 16 | 10 | 0.847 | 0.748 | 0.787 | 0.636 | 0.863 | 0.802 |
| usps | 11000 | 256 | 10 | 0.778 | 0.453 | 0.676 | 0.293 | 0.756 | 0.368 |
| letters | 20000 | 16 | 26 | 0.478 | 0.350 | 0.224 | 0.124 | 0.380 | 0.278 |
| mnist | 70000 | 784 | 10 | 0.747 | 0.420 | 0.677 | 0.187 | 0.747 | 0.260 |
| skin | 245057 | 3 | 2 | 0.767 | NA | 0.878 | NA | 0.953 | NA |
| covertype | 581012 | 54 | 7 | 0.218 | NA | 0.167 | NA | 0.262 | NA |

## H    PARAMETER SETTING

**Datasets**: spiral Chang and Yeung (2008), 4C Zuo and Hou (2022), AC Marin et al. (2019), RingG Ting et al. (2023), complex9 Salvador and Chan (2004), cure-t2-4k Van Craenendonck and Blockeel (2017), landsat Wang et al. (2023), spambase Hopkins and Suermondt (1999), waveform3 Wang et al. (2023), pendigits Alpaydin and Alimoglu (1996), usps Cai et al. (2010), letters Frey and Slate (1991), mnist Cai et al. (2011), skin Bhatt and Dhall (2009), covertype Blackard and Dean (1999).

In order to reduce the contingency introduced by hyper-parameter settings, we tried multiple hyper-parameters for each algorithm and reported the best results Xie et al. (2016); Saha et al. (2023); Nie et al. (2024); Tang et al. (2022). The hyper-parameters of the algorithms used in the experiments are shown in Table 7.

The inability to automatically determine parameters is a limitation of our algorithm, and we leave this as future work. Since NMI requires labels, in practice, one approach is to use label-independent metrics such as CH (Calinski-Harabasz)Caliński and Harabasz (1974) to select hyperparameters. Figure 10 shows the results of CH and NMI. CH and NMI have similar distributions: parameters with higher CH also tend to have higher NMI, and parameters with higher NMI also tend to have higher CH, especially when $\psi \geq 32$. Therefore, in practice, we can use unsupervised methods such as CH to select parameters. We also noticed that DSPEC has better results in the band area where $\psi$ ranges from 32 to 512 and $\tau$ ranges from 0.3-0.5 ($\psi$=32) to 0.1-0.3 ($\psi$=512).

Table 7: Parameter search ranges.

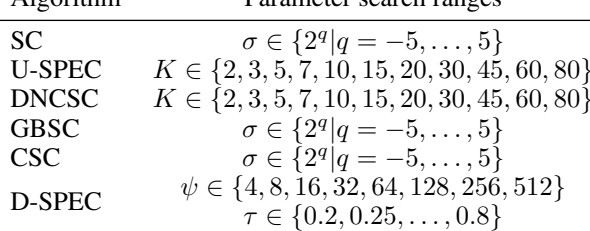

| Algorithm | Parameter search ranges |
|---|---|
| SC | $\sigma \in \{2^q | q = -5, \ldots, 5\}$ |
| U-SPEC | $K \in \{2, 3, 5, 7, 10, 15, 20, 30, 45, 60, 80\}$ |
| DNCSC | $K \in \{2, 3, 5, 7, 10, 15, 20, 30, 45, 60, 80\}$ |
| GBSC | $\sigma \in \{2^q | q = -5, \ldots, 5\}$ |
| CSC | $\sigma \in \{2^q | q = -5, \ldots, 5\}$ |
| D-SPEC | $\psi \in \{4, 8, 16, 32, 64, 128, 256, 512\}$ $\tau \in \{0.2, 0.25, \ldots, 0.8\}$ |

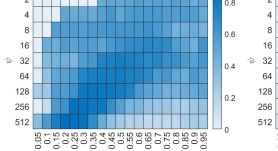 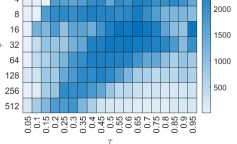 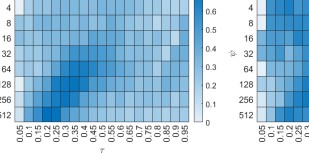 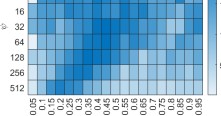

| (a) NMI on Pendigits | (b) CH on Pendigits | (c) NMI on Mnist | (d) CH on Mnist |
|---|---|---|---|

Figure 10: The heatmap of CH and NMI on Pendigits and Mnist.

# I   RESULTS OF THE FIVE LARGEST DATASETS

The results in terms of NMI, ARI, F-measure and runtime on the five largest datasets are shown in Table 8, Table 9, Table 10, Table 11, respectively.

Table 8: Average NMI scores over 10 runs. The best score in each dataset is highlighted in bold.

| dataset | #n | #d | k | SC | U-SPEC | DNCSC | FastSC | GBSC | D-SPEC |
|---|---|---|---|---|---|---|---|---|---|
| TB1M | 1M | 2 | 2 | N/A | 0.957 | 0.970 | 0.001 | O/M | **0.974** |
| SF2M | 2M | 2 | 4 | N/A | 0.799 | 0.818 | 0.001 | O/M | **0.936** |
| CC5M | 5M | 2 | 3 | N/A | **0.999** | 0.998 | N/A | O/M | **0.999** |
| CG10M | 10M | 2 | 11 | N/A | 0.809 | 0.841 | N/A | O/M | **0.942** |
| Flower20M | 20M | 2 | 13 | N/A | 0.892 | O/M | N/A | O/M | **0.963** |

Table 9: Average ARI scores over 10 runs. The best score in each dataset is highlighted in bold.

| dataset | #n | #d | k | SC | U-SPEC | DNCSC | FastSC | GBSC | D-SPEC |
|---|---|---|---|---|---|---|---|---|---|
| TB1M | 1M | 2 | 2 | N/A | 0.981 | 0.988 | 0.001 | O/M | **0.989** |
| SF2M | 2M | 2 | 4 | N/A | 0.748 | 0.903 | 0.001 | O/M | **0.966** |
| CC5M | 5M | 2 | 3 | N/A | **1.000** | 0.999 | N/A | O/M | 0.999 |
| CG10M | 10M | 2 | 11 | N/A | 0.525 | 0.913 | N/A | O/M | **0.959** |
| Flower20M | 20M | 2 | 13 | N/A | 0.811 | O/M | N/A | O/M | **0.966** |

# J   FOUNDMENTAL LIMITATIONS OF SPECTRAL CLUSTERING

The results in terms of NMI, ARI, and F-measure on the three datasets are shown in Table 12, Table 13, and Table 14 respectively.

Table 10: Average F-measure scores over 10 runs. The best score in each dataset is highlighted in bold.

| dataset | #n | #d | k | SC | U-SPEC | DNCSC | FastSC | GBSC | D-SPEC |
|---|---|---|---|---|---|---|---|---|---|
| TB1M | 1M | 2 | 2 | N/A | 0.995 | **0.997** | 0.509 | O/M | **0.997** |
| SF2M | 2M | 2 | 4 | N/A | 0.735 | 0.739 | 0.243 | O/M | **0.966** |
| CC5M | 5M | 2 | 3 | N/A | **1.000** | 0.999 | N/A | O/M | **1.000** |
| CG10M | 10M | 2 | 11 | N/A | 0.586 | 0.802 | N/A | O/M | **0.974** |
| Flower20M | 20M | 2 | 13 | N/A | 0.836 | O/M | N/A | O/M | **0.974** |

Table 11: Runtime (seconds) over 10 runs. The best score in each dataset is highlighted in bold.

| dataset | #n | #d | k | SC | U-SPEC | DNCSC | FastSC | GBSC | D-SPEC |
|---|---|---|---|---|---|---|---|---|---|
| TB1M | 1M | 2 | 2 | N/A | 17.512 | **5.179** | 57.982 | O/M | 15.920 |
| SF2M | 2M | 2 | 4 | N/A | 53.107 | **13.201** | 274.534 | O/M | 49.310 |
| SF5M | 5M | 2 | 3 | N/A | 74.690 | **20.174** | N/A | O/M | 98.572 |
| SF10M | 10M | 2 | 11 | N/A | 625.271 | **245.621** | N/A | O/M | 545.864 |
| SF20M | 20M | 2 | 13 | N/A | 1258.085 | O/M | N/A | O/M | **1245.407** |

Table 12: Average NMI scores over 10 runs. The best score in each dataset is highlighted in bold.

| dataset | #n | #d | k | SC | USPE | DNCSC | FastSC | GBSC | D-SPEC |
|---|---|---|---|---|---|---|---|---|---|
| DSSS | 900 | 2 | 3 | 0.909 | 0.848 | 0.917 | 0.737 | 0.630 | **0.937** |
| DSDS | 900 | 2 | 3 | 0.797 | 0.752 | 0.787 | 0.559 | 0.525 | **0.899** |
| OGOL | 1400 | 2 | 2 | 0.454 | 0.878 | 0.515 | 0.595 | 0.538 | **0.885** |

Table 13: Average ARI scores over 10 runs. The best score in each dataset is highlighted in bold.

| dataset | #n | #d | k | SC | USPE | DNCSC | FastSC | GBSC | D-SPEC |
|---|---|---|---|---|---|---|---|---|---|
| DSSS | 900 | 2 | 3 | 0.937 | 0.819 | 0.945 | 0.604 | 0.479 | **0.966** |
| DSDS | 900 | 2 | 3 | 0.908 | 0.814 | 0.899 | 0.540 | 0.348 | **0.973** |
| OGOL | 1400 | 2 | 2 | 0.409 | 0.922 | 0.497 | 0.597 | 0.530 | **0.936** |

Table 14: Average F-measure scores over 10 runs. The best score in each dataset is highlighted in bold.

| dataset | #n | #d | k | SC | USPE | DNCSC | FastSC | GBSC | D-SPEC |
|---|---|---|---|---|---|---|---|---|---|
| DSSS | 900 | 2 | 3 | 0.978 | 0.850 | 0.981 | 0.608 | 0.524 | **0.987** |
| DSDS | 900 | 2 | 3 | 0.547 | 0.540 | 0.597 | 0.632 | 0.601 | **0.964** |
| OGOL | 1400 | 2 | 2 | 0.813 | 0.980 | 0.849 | 0.737 | 0.861 | **0.981** |

## K    IMAGE SEGMENTATION

We show an example that use D-SPEC to cluster images for image segmentation. We first convert the images (shown in Figure 11(a)) from sRGB values to CIE 1976 Lab* values. (`https://en.wikipedia.org/wiki/CIELAB_color_space`). Then we cluster the images in LAB space. The segmented results are shown in 11(b-d).

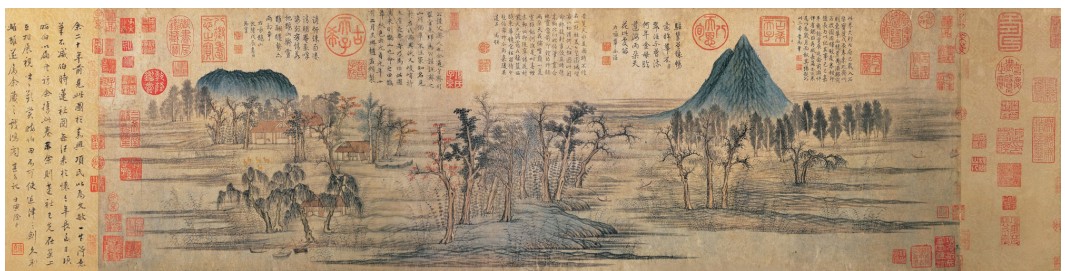

(a) Image

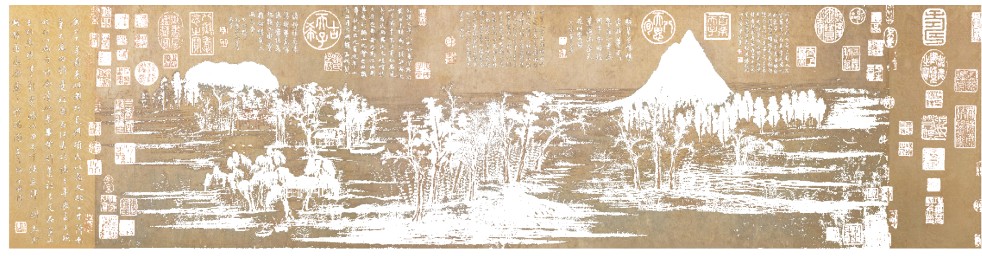

(b) Cluster 1

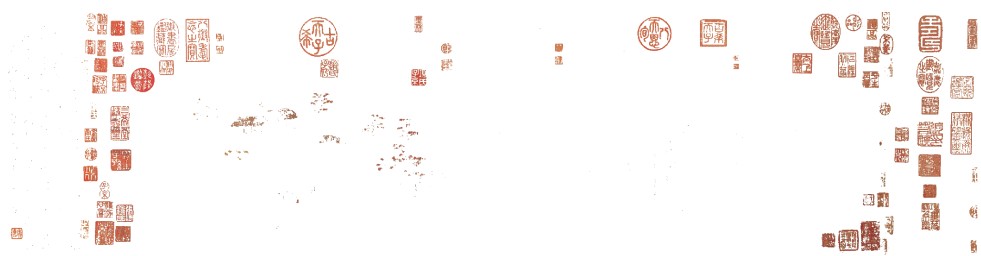

(c) Cluster 2

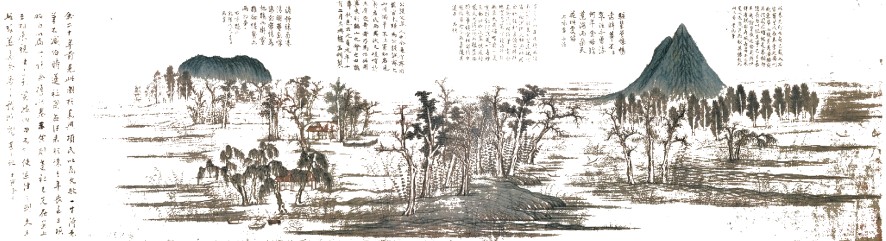

(d) Cluster 3

Figure 11: Image Segmentation of D-SPEC.

