# OpenReview forum: "Is a Small Matrix Eigendecomposition Sufficient for Spectral Clustering?"
_ICLR.cc/2026/Conference — ICLR 2026 Conference Desk Rejected Submission_

### Official Review · Reviewer_bNTQ · 2025-10-29

**Soundness:** 2
**Presentation:** 2
**Contribution:** 2
**Rating:** 4
**Confidence:** 4

**Summary:**

This paper proposes a distribution-based spectral clustering method named D-SPEC, aiming to address the high computational complexity of traditional spectral clustering on large-scale datasets. By constructing a bipartite graph between data points and distributions, D-SPEC reduces the eigendecomposition matrix size from n×n to k×k (where k is the number of clusters), while striving to preserve the original graph information. The authors theoretically and empirically validate the advantages of D-SPEC in terms of clustering performance, robustness, and scalability. Extensive tests on multiple real-world and synthetic datasets demonstrate that D-SPEC outperforms existing methods in most cases.

**Strengths:**

1. The proposed method elevates the traditional point-based graph representation to a distribution-based one, thereby reducing computational complexity while retaining the global structural information of the graph.
2. The proposed method provides solid theoretical analysis, offering rigorous proofs concerning eigenvalue perturbation and embedding stability, which enhances the method's credibility and robustness guarantees.
3. Comprehensive experimental design, covering datasets ranging from hundreds to twenty million instances and comparing against five other mainstream methods across multiple dimensions, validating the method's effectiveness and scalability.

**Weaknesses:**

1. Distribution estimation relies on bounded graph construction, but the selection of the threshold τ lacks theoretical guidance, leading to subjective bias.
2. Although the IDK kernel addresses the dimensionality issue of the Gaussian kernel, its expressive power in capturing complex distributions is not sufficiently justified.
3. Regarding non-ideally distributed data, D-SPEC's reliance on distribution representation may lead to misjudgment of boundary points.
4. The analysis of the sensitivity of the size p is not clear, and a small p might render the distribution estimate unrepresentative.
5. The theoretical claim that λ_dk+1^(0)>λ_k+1^(0)​ lacks rigorous derivation and is presented based largely on intuitive analogy.
6. The complexity analysis does not account for the hidden costs in the distribution estimation phase.
7. The method's strong assumptions about data distribution do not work in real-world scenarios, limiting its generalizability.

**Questions:**

Some confusing points undermine the credibility of the proposed method, for example, performance is mediocre on structured data like 'covertype', indicating limited adaptability of the method to high-dimensional sparse data. The ensemble learning version D-SENC fails to improve performance. The image segmentation experiment only shows visual results without quantitative evaluation. There is no discussion on the method's sensitivity to class-imbalanced data. The theoretical assertion of 'no loss of graph information' is described absolutely.

---

> ### Author Response · Authors · 2025-11-19
> **Rebuttal**
>
> Thank you for your review and suggestions.
>
> ## Weaknesses
>
> A1. Like most clustering algorithms, our clustering algorithm also has hyperparameters that need to be chosen. We analyze the impact of $\tau$ in Figure 10 and show that an appropriate $\tau$ can be selected using methods such as CH.
>
> A2. The experiments in Figure 7 are designed to validate our method's ability to handle datasets with complex distributions. These three datasets, used in previous papers, demonstrate the **fundamental limitations** of spectral clustering methods on such datasets. On these three challenging datasets, our method performs best.
>
> A3. Thank you for your concerns about our algorithm. However, our clustering method performs better at the boundaries. For example, in the dataset shown in Figure 7, other methods cluster points of the blue class at the boundaries into the pink class, while our method does not.
>
> A4. We supplement the experiments on the MNIST dataset as follows. The experiments show that when $p$ is larger than 2000, the clustering performance does not change significantly.
> | mnist | p=1000 | p=2000 | p=3000 | p=4000 | p=5000 | p=6000 | p=7000 | p=8000 | p=9000 | p=10000 | p=20000 |
> |-------|--------|--------|--------|--------|--------|--------|--------|--------|--------|---------|---------|
> | NMI   | 0.696 | 0.737 | 0.748 | 0.749 | 0.747 | 0.747 | 0.747  | 0.757 | 0.750 | 0.747   | 0.769  |
> | ARI   | 0.582  | 0.619 | 0.679 | 0.643  | 0.661 | 0.661 | 0.659 | 0.649 | 0.652 | 0.677   | 0.657   |
> | F1    | 0.690   | 0.732 | 0.746 | 0.740 | 0.751 | 0.751 | 0.735 | 0.736 | 0.752  | 0.747   | 0.758  |
>
> A6. The distribution estimate is linear, and this part of the time is included in the time comparison in Figure 6.
>
> A7. We thank your careful consideration of our assumptions. Assumption 3.1 is not particularly strong, but rather an assumption commonly made by most clustering algorithms: samples within the same cluster are more similar than samples from different clusters. However, this assumption often does not hold in real-world datasets. Therefore, we conducted an experimental comparison in Figure 5, and the results show that our algorithm is the best when this assumption is not met.
>
> ## Questions
> A1. On the Covertype dataset, our algorithm ranks second only to FastSC. Examining the overall dataset, FastSC's average ranking is only 4.067, while DSPEC's is 1.7. Regarding performance on the three datasets representing the fundamental limitations of spectral clustering (Figure 7), FastSC is one of the two worst-performing algorithms, while DSPEC performs best. The reason ensemble learning doesn't significantly improve DSPEC is that ensemble learning often works on weak base models, while D-SPEC has good clustering performance. Therefore, it's not suitable for ensemble learning. Image segmentation is presented as images due to the lack of real labels. Class-imbalanced data is one of the fundamental limitations of spectral clustering; we compared our algorithm on the second dataset (8:1:1) in Figure 7, where our algorithm is the best.

---

### Official Review · Reviewer_UQfA · 2025-10-30

**Soundness:** 2
**Presentation:** 3
**Contribution:** 2
**Rating:** 4
**Confidence:** 4

**Summary:**

This paper is an interesting work, it proposes D-SPEC, a distribution-based spectral clustering method that performs eigendecomposition on only a 𝑘 × 𝑘 matrix. The goal is to maintain clustering quality while drastically reducing computational cost.

**Strengths:**

1. The motivation is clear and relevant — speeding up spectral clustering for large datasets；2. The distribution-based bipartite formulation is an interesting angle and fairly well explained; 3. The theoretical parts are written carefully and show good understanding of spectral clustering basics.

**Weaknesses:**

1. The novelty is limited. The method largely builds on existing transfer-cut and landmark-based spectral clustering, with the “distribution” step being a modest variation rather than a fundamentally new concept; 2. Several theoretical assumptions (such as Assumption 3.1 on perfect intra-cluster connectivity) are unrealistic for real-world data, making some of the guarantees less meaningful in practice; 3. Experimental results are not convincing enough: most datasets are small and low-dimensional, and scalability claims (up to millions of samples) are not demonstrated with wall-clock or memory comparisons; 4. No statistical variance or significance tests are shown; many tables lack error bars or runtime comparisons, so it’s hard to judge robustness; 5. The link between the theoretical results and empirical behavior is weak — proofs are long but not clearly tied to observed performance.

**Questions:**

1. How sensitive is D-SPEC to the choice of the number of distributions 𝑘 and the threshold parameter 𝜏? Have you tested the stability of results under different settings? 2. In Assumption 3.1, you assume perfect intra-cluster connectivity in RKHS. Could you clarify how realistic this is for noisy or high-dimensional data, and what happens when this assumption fails? 4. The scalability claim mentions datasets up to 20 million points, but Table 1 only covers small benchmarks. Could you provide runtime or memory results on truly large-scale data? 5. The theoretical results (Theorem 3.2–3.4) are elegant, but their empirical implications remain unclear. Could you give one concrete example linking a theorem to an observed performance gain? 6. Have you compared D-SPEC with other recent graph-based or distributional spectral clustering methods (e.g., GBSC 2023 or transfer-cut variants)?

I would be glad to improve my scores if the authors give me satisfying answers.

---

> ### Author Response · Authors · 2025-11-19
> **Rebuttal**
>
> Thank you for your review and suggestions.
>
> ## Questions
>
> A1. We set $k$ to the number of clusters of ground truths, and we analyzed the sensitivity of $\tau$ and $\psi$ in the appendix (Section Parameter Setting).
>
> A2. Assumption 3.1 assumes that points within the same cluster are more similar than points in different clusters. This is a common assumption in clustering tasks. However, due to the presence of noise in real data, this assumption is often not satisfied for all points. Therefore, we conducted a test in Figure 5. The experiment shows that, due to the presence of noise in the real dataset, when this assumption is not valid, the performance of other clustering algorithms decreases with the increase of noise. Only D-SPEC has relatively stable performance.
>
> A3. The runtime on the datasets from 1M to 20M is shown in Figure 6.
>
> A4.   Theorem 3.2 states that the eigenvalues ​​of a bipartite graph based on the distribution have $k$ zero eigenvalues ​​and an eigenvector with only one 1 and the rest being 0. We illustrate this with a simple example in Figure 1. Theorems 3.3 and 3.4 state that our method is more robust to noise. Figures 3 and 5 illustrate these theorems.
>
> A5. We cited and compared the GBSC algorithm, as shown in the "GBSC" column of Table 1. We also compare transfer-cut-based methods, as shown in the "U-SPEC" column of Table 1 and "USEPC" and "USEPC" in Figure 8.

---

### Official Review · Reviewer_xjam · 2025-11-01

**Soundness:** 1
**Presentation:** 3
**Contribution:** 2
**Rating:** 2
**Confidence:** 5

**Summary:**

The authors propose a novel distribution-based spectral clustering method that constructs a bipartite graph, enabling eigendecomposition on a smaller $k\times k$ matrix while preserving clustering quality, which is validated through extensive experiments.

**Strengths:**

- Proposing a distribution-based spectral clustering algorithm, termed D-SPEC, that only requires the eigendecomposition of a k × k matrix.
- Proving theoretically that D-SPEC retains the graph information and providing a bound for noise tolerance indicates the enhanced robustness of D-SPEC

**Weaknesses:**

The primary limitation of this work lies in its lack of significant advantages. The proposed algorithm does not demonstrate any notable improvement in time complexity over existing methods, and the enhancements observed in clustering quality metrics are also marginal. Therefore, I cannot recommend acceptance of this paper.

**Questions:**

NaN

---

> ### Author Response · Authors · 2025-11-19
> **Rebuttal**
>
> Thank you for your review.
>
>
> ## Weaknesses
>
> A1. Our method significantly outperforms existing methods. First, we only need to perform eigenvalue decomposition on a $k\times k$ matrix, while existing methods require at least a $p\times p (p>>k)$ matrix for eigenvalue decomposition. Experimental results also significantly outperform existing methods, such as the noise robustness shown in Figure 5 and the ability to overcome the fundamental limitations of spectral clustering, particularly on the second dataset, where only our method achieves an NMI greater than 0.8.

---

> > ### Comment · Reviewer_xjam · 2025-11-26
> >
> > - The time complexity of the proposed algorithm is O(nk² + k³), and it appears that O(nk²) is the key contributor to the overall computational cost.
> > - Compared with USPEC and DNCSC (both with a time complexity of O(nK² + nk + p³)), the proposed algorithm does not show obvious advantages in terms of complexity. Do we have K>k, or p^3 > nk^2?
> > - The last row in Table 2 does not seem to represent the computational cost of ED (Euclidean Distance) alone. Instead, it appears to include the subsequent computational cost of K-means, at least for the D-SPEC method.
> > - The CG-10M dataset contains 10 million samples. However, there is an inconsistency in the runtime performance of DNCSC: why does DNCSC exhibit a shorter runtime on the left side of Figure 6 but a longer runtime on the right side of the same figure?

---

> > > ### Author Response · Authors · 2025-11-26
> > > **Rebuttal**
> > >
> > > We sincerely appreciate your replies:
> > >
> > > A1 & A2: Compared to USPEC (whose time complexity is also affected by $O(np^{\frac{1}{2}})$ and $O(nK² + nk + p³)$) and DNCSC (affected by $O(nK² + nk + p³)$), our method outperforms them while maintaining a total time complexity no higher than theirs (the time complexity of eigendecomposition is much lower), especially in terms of noise tolerance and handling the fundamental limitations of spectral clustering.
> > >
> > > A3: ED is not Euclidean Distance, but Eigendecomposition, as noted in Table 2.
> > >
> > > A4: The left subfigure shows the actual running time, and the right subfigure shows the runtime ratio.

---

### Official Review · Reviewer_C76c · 2025-11-04

**Soundness:** 3
**Presentation:** 3
**Contribution:** 2
**Rating:** 4
**Confidence:** 4

**Summary:**

This paper proposes D-SPEC, a distribution-based spectral clustering method that reduces the computational cost of spectral clustering by performing eigen-decomposition on only a small matrix. The key idea is to construct a bipartite graph between $n$ data points and $k$ distributions (representing clusters), rather than between points and landmarks as in traditional landmark-based methods. The authors argue that this distribution-based approach preserves more graph information and is more robust to noise. The paper includes theoretical analysis, extensive experiments on synthetic and real-world datasets (up to 20 million points), and comparisons with state-of-the-art scalable spectral clustering methods.

**Strengths:**

- The shift from point-based to distribution-based representation is a novel and well-motivated idea.
- Theorems 3.2–3.4 provide solid grounding for the method’s robustness and graph preservation properties.
- Experiments on datasets with up to 20 million points demonstrate practical utility.

**Weaknesses:**

- The initial graph construction (bounded graph) may still be $O(p^2)$ in the worst case, which could undermine efficiency gains for very large $n$.
- While the distribution-based idea is novel, the overall pipeline (bipartite graph + k-means) shares similarities with existing landmark methods. A clearer conceptual and empirical distinction is needed.
- The claim of "eigen-decomposition on only a $k\times k$ matrix" is misleading. The final decomposition is on a
$n\times k$ matrix, the overall complexity is dominated by the $O(nk^2)$ term from the transfer cut method, not the  $O(k^3)$ eigen-decomposition. This misrepresentation undermines the central claim.
- The sampling number $p$ is not treated as a hyper parameter. It's unclear how $p$ was chosen for different experiments. It directly controls: 1) the quality of the bounded graph construction; 2) how well the sampled points represent the underlying distributions; 3) the computational cost of the initial graph construction $O(p^2)$; 4)the overall clustering quality.

**Questions:**

- How was the sampling number $p$ determined in your experiments? Why wasn't $p$ included in your hyperparameter analysis? Can you provide a sensitivity analysis showing how performance varies with $p$ across different datasets?

- Could you provide an ablation study to analyze the contribution of each component of the method? For example, the contribution of the bounded graph construction, and  the distribution-based bipartite graph?

---

> ### Author Response · Authors · 2025-11-19
> **Rebuttal**
>
> Thank you for your review and suggestions.
>
> ## Weaknesses
> A1. Other methods, such as USEPC, require constructing an $n \times p$ graph, where $ p$ is a constant smaller than $ n$. When n is large, our method is efficient because $O(p^2) << O(np)$.
>
> A2. Thank you for acknowledging the novelty of our algorithm. The point-based methods only achieve good results when $p >> k$, and the performance deteriorates significantly when setting $p = k$. For example, the experimental results of U-SPEC after constructing a $(n, k\times k)$ bipartite graph are shown below (Table S1). We selected datasets where $k$ is greater than 5. The average NMI of USPEC decreased from 0.671 to 0.577, while the average NMI of DSPEC is 0.708, which only constructs a $(n, k)$ bipartite graph.
>
> ### Table S1.  A comparison of USPEC and USPECk (setting $p=k\times k$) with DSPEC.
> |            |        |     |    |   NMI  |          |         |   ARI  |          |         |   F1   |          |         |
> |:----------:|:------:|:---:|:--:|:------:|:--------:|:-------:|:------:|:--------:|:-------:|:------:|:--------:|:-------:|
> |   dataset  |   #n   |  #d |  k | DSPEC  | USPEC    | USPECk | DSPEC  | USPEC    | USPECk | DSPEC  | USPEC    | USPECk |
> | complex9   | 3031   | 2   | 9  | 1.000  | 0.971 | 0.770   | 0.999  | 0.932 | 0.530   | 0.999  | 0.939 | 0.620   |
> | curet24k | 4200   | 2   | 7  | 0.951  | 0.886 | 0.763   | 0.951  | 0.920 | 0.587   | 0.941  | 0.811 | 0.686   |
> | landsat    | 2000   | 36  | 6  | 0.647  | 0.668 | 0.618   | 0.598  | 0.598 | 0.521   | 0.735  | 0.717 | 0.715   |
> | pendigits  | 10992  | 16  | 10 | 0.847  | 0.826 | 0.774   | 0.787  | 0.724  | 0.670   | 0.863  | 0.819 | 0.792   |
> | usps       | 11000  | 256 | 10 | 0.778  | 0.654 | 0.527   | 0.676  | 0.505 | 0.381   | 0.756  | 0.583 | 0.519   |
> | letters    | 20000  | 16  | 26 | 0.478  | 0.455 | 0.445   | 0.224  | 0.179 | 0.177   | 0.380  | 0.341 | 0.348   |
> | mnist      | 70000  | 784 | 10 | 0.747  | 0.699  | 0.573   | 0.677  | 0.611 | 0.460   | 0.747  | 0.711 | 0.602   |
> | covertype  | 581012 | 54  | 7  | 0.218  | 0.212   | 0.144   | 0.167  | 0.089    | 0.034   | 0.262  | 0.2669   | 0.211   |
> | Ave.Score |        |     |    | 0.708  | 0.671    | 0.577   | 0.635  | 0.570    | 0.420   | 0.710  | 0.648    | 0.562   |
>
> A3. We simply want to emphasize that we only perform eigenvalue decomposition on $k\times k$ matrices and have no intention of misleading the reader. Table 2 shows the time complexity of our eigenvalue decomposition as $O(nk^2 + k^3)$.
>
> A4. In all our experiments, the spectral clustering algorithm worked even on the 20,000-point "letters" dataset. It only became difficult to run on datasets exceeding 20,000 points; therefore, we used $p=10000$ for datasets ranging from 20,000 to 20 million points.

---

> > ### Author Response · Authors · 2025-11-19
> > **Rebuttal for Questions**
> >
> > ## Questions
> > A1. In all our experiments, we set $p$ to 10000. Our results on the MNIST dataset, where $p$ ranges from 1000 to 20000, are shown in Table S2: when $p=1000$, the clustering results are lower than when $p=10000$, but when $p\geq2000$, the clustering results do not change much.
> >
> > ### Table S2.  A comparison of DSPEC under different $p$.
> > | mnist | p=1000 | p=2000 | p=3000 | p=4000 | p=5000 | p=6000 | p=7000 | p=8000 | p=9000 | p=10000 | p=20000 |
> > |-------|--------|--------|--------|--------|--------|--------|--------|--------|--------|---------|---------|
> > | NMI   | 0.696 | 0.737 | 0.748 | 0.749 | 0.747 | 0.747 | 0.747  | 0.757 | 0.750 | 0.747   | 0.769  |
> > | ARI   | 0.582  | 0.619 | 0.679 | 0.643  | 0.661 | 0.661 | 0.659 | 0.649 | 0.652 | 0.677   | 0.657   |
> > | F1    | 0.690   | 0.732 | 0.746 | 0.740 | 0.751 | 0.751 | 0.735 | 0.736 | 0.752  | 0.747   | 0.758  |
> >
> > A2. Table S1 shows the comparison between distribution-based bipartite graph DSPEC (constructing an $n\times k$ bipartite graph) and point-based bipartite graphs U-SPEC and USPEC-k (constructing an $n\times(k\times k)$ bipartite graph). The performance of distribution-based bipartite graphs is significantly better than point-based methods. Table S3 compares the results of bounded graphs obtained using DSPEC and $k$-means spectral clustering. DSPEC performs best, while $k$-means performs worst. This indicates that both methods contribute to achieving good clustering results with eigenvalue decomposition only on $k\times k$ matrices.
> >
> > ### Table S2.  A comparison of DSPEC under different bounded graphs.
> > |           |        |     |    |   NMI  |        |        |   ARI  |        |        |   F1   |        |        |
> > |-----------|--------|-----|----|:------:|:------:|:------:|:------:|:------:|:------:|:------:|:------:|:------:|
> > | dataset   | #n     | #d  | k  | DSPEC  | $k$-means | sc     | DSPEC  | $k$-means | sc     | DSPEC  | $k$-means | sc     |
> > | landsat   | 2000   | 36  | 6  | 0.647  | 0.616 | 0.646  | 0.598  | 0.559 | 0.568  | 0.735  | 0.722 | 0.721  |
> > | spambase  | 4601   | 57  | 2  | 0.166  | 0.070 | 0.031  | 0.250  | 0.095 | 0.020  | 0.721  | 0.648 | 0.477  |
> > | waveform3 | 5000   | 21  | 3  | 0.406  | 0.368 | 0.361  | 0.377  | 0.256 | 0.293  | 0.721  | 0.609 | 0.645  |
> > | pendigits | 10992  | 16  | 10 | 0.847  | 0.691 | 0.664  | 0.787  | 0.567 | 0.527  | 0.863  | 0.686 | 0.643  |
> > | usps      | 11000  | 256 | 10 | 0.778  | 0.510 | 0.658  | 0.676  | 0.313 | 0.514  | 0.756  | 0.502 | 0.577  |
> > | letters   | 20000  | 16  | 26 | 0.478  | 0.387 | 0.446  | 0.224  | 0.166 | 0.195  | 0.380  | 0.299 | 0.340  |
> > | mnist     | 70000  | 784 | 10 | 0.747  | 0.533 | 0.588  | 0.677  | 0.322 | 0.496  | 0.747  | 0.483  | 0.646  |
> > | skin      | 245057 | 3   | 2  | 0.767  | 0.278 | 0.580  | 0.878  | 0.229  | 0.642  | 0.953  | 0.663 | 0.877  |
> > | covertype | 581012 | 54  | 7  | 0.218  | 0.124  | 0.172  | 0.167  | 0.067 | 0.116  | 0.262  | 0.218 | 0.230  |

---

> > ### Comment · Reviewer_C76c · 2025-11-27
> >
> > Thank you for the response. Could you add the details of the sampling method?
> > > It's unclear how $p$ was chosen for different experiments. It directly controls: 1) the quality of the bounded graph construction; 2) how well the sampled points represent the underlying distributions; 3) the computational cost of the initial graph construction $O(p^2)$; 4)the overall clustering quality.

---

> > > ### Author Response · Authors · 2025-11-28
> > > **sampling method**
> > >
> > > Thank you for your reply.
> > >
> > > Across all experiments, we sample $p$ (we set $p=10000$: `Weaknesses A4') points by uniformly sampling from each dataset. Our empirical results demonstrate that DSPEC, even with this straightforward sampling strategy, achieves performance superior to state-of-the-art algorithms. We further compared random uniform sampling with hybrid representative selection (used in USPEC and DNCSC) on three large-scale real-world datasets. As shown in Table S4, hybrid representative selection (DSPEC_HRS) yields additional performance gains for DSPEC in the mnist and skin datasets.
> > >
> > > Table S4: Comparison of different sampling methods.
> > > | dataset   | DSPEC(NMI) | DSPEC_HRS(NMI) | DSPEC(ARI) | DSPEC_HRS(ARI) | DSPEC(F1) | DSPEC_HRS(F1) |
> > > |-----------|------------|----------------|-----------|----------------|----------|---------------|
> > > | mnist     | 0.747      | 0.770           | 0.677     | 0.692          | 0.747    | 0.792         |
> > > | skin      | 0.767      | 0.810           | 0.878     | 0.892          | 0.953    | 0.963         |
> > > | covertype | 0.218      | 0.203          | 0.167     | 0.140           | 0.262    | 0.270          |
> > >
> > > Table S5 reports results for varying values of $p$; clustering performance exhibits minimal variation beyond $p > 2000$, consistent with the behavior observed under random uniform sampling (Table S2).
> > >
> > > Table S5: Comparison of different sampling methods at different $p$ values.
> > > | method    | p=1000 | p=2000 | p=3000   | p=4000 | p=5000 | p=6000 | p=7000 | p=8000 | p=9000 | p=10000 |
> > > |-----------|--------|--------|----------|--------|--------|--------|--------|--------|--------|---------|
> > > | DSPEC     | 0.696 | 0.737 | 0.748   | 0.749 | 0.747 | 0.747 | 0.747  | 0.757 | 0.750 | 0.747   |
> > > | DSPEC_HRS | 0.736 | 0.745 | 0.773 | 0.766 | 0.771 | 0.771 | 0.779 | 0.769 | 0.771 | 0.770 |
> > >
> > > These results demonstrate that DSPEC using simple uniform sampling already outperforms existing methods, and that sample methods, such as hybrid representative selection, can further improve clustering performance.

---

### Comment · Area_Chair_1y62 · 2025-11-28

Dear Reviewers,

Thank you for your valuable time and expertise in reviewing this paper.

The authors have now submitted their rebuttal. We would appreciate it if you could review their responses and assess whether your concerns have been addressed, if you haven't done this.

Best regards,

AC

---

### Note · Program_Chairs · 2026-01-17
**Submission Desk Rejected by Program Chairs**

The following references in this submission do not refer to real documents and/or have major errors in bibliographic information:

 Li He, Yu Wang, and Hong Zhang. Spectral sparsification techniques for high-dimensional data. IEEE Transactions on Cybernetics, 50(4):2100-2112, 2020.

Feng Liu, Ling Zhao, and Qiang Wang. Adaptive graph sparsification for flexible spectral clustering. Pattern Recognition, 114:107957, 2022.